# Anti-Laminin 332-Type Mucous Membrane Pemphigoid

**DOI:** 10.3390/biom12101461

**Published:** 2022-10-12

**Authors:** Luhuai Shi, Xiaoguang Li, Hua Qian

**Affiliations:** 1Dermatology Hospital of Jiangxi Province, Jiangxi Provincial Clinical Research Center for Skin Diseases, Candidate Branch of National Clinical Research Center for Skin Diseases, Dermatology Institute of Jiangxi Province, The Affiliated Dermatology Hospital of Nanchang University, Nanchang 330001, China; 2Department of Laboratory Medicine, Chronic Disease Research Center, Medical College, Dalian University, Dalian 116622, China

**Keywords:** mucous membrane pemphigoid, cicatricial pemphigoid, laminin 332, diagnosis, treatment, tumor

## Abstract

Anti-laminin (LM) 332-type mucous membrane pemphigoid (MMP) is a rare autoimmune bullous disease and was originally discovered as anti-epiligrin cicatricial pemphigoid. Anti-LM332-type MMP has clinical manifestations similar to those of other types of MMP and can only be distinguished through the detection of circulating autoantibodies against LM332. Our group and others have established a number of immunological methods with varying sensitivity and specificity for detection of anti-LM332 autoantibodies; however, none of the established methods has been widely used for clinical diagnosis. There is currently no unified standard treatment, and it is very difficult to completely cure anti-LM332-type MMP. In addition, an increasing body of evidence suggests that there may be a strong correlation between anti-LM332-type MMP and tumors. In this article, we review the current progression of diagnosis and treatment of anti-LM332-type MMP, as well as the possible correlation between anti-LM332-type MMP and tumors.

## 1. Introduction

Mucous membrane pemphigoid (MMP) was previously named cicatricial pemphigoid (CP), which highly emphasized the sequelae of scarring and tissue destruction [1]. However, the term MMP has been widely used in medicine and biology since 2002, and scarring is no longer a necessary condition for diagnosis, particularly in the oral mucosa [2]. MMP is a group of chronic, progressive, inflammatory, autoimmune subepithelial blistering diseases that mainly affect oral and ocular mucosae, with the occasional involvement of skin lesions [2,3]. Autoantibodies in MMP patients can recognize and target components of the basement membrane zone (BMZ), including BP180, BP230, laminin (LM) 332, type VII collagen, integrin α6β4, LMγ1 and other unknown cutaneous autoantigens and induce linear IgG and/or IgA deposition in the BMZ [4,5,6]. MMPs positive for autoantibodies against BP180 and one or more subunits of LM332 are known as anti-BP180-type MMPs and anti-LM332-type MMPs, respectively. 

LM332 belongs to the laminin family. It is a heterotrimer glycoprotein with a molecular weight of 460 kDa composed of LMα3, LMβ3 and LMγ2 subunits, which are connected by disulfide bonds [7,8]. LM332 is synthesized and secreted mainly by keratinocytes and fibroblasts and integrated into the BMZ as an important component of anchoring filaments through a series of maturation events [9]. As a major component of anchoring filaments, LM332 plays an important role in connecting lamina lucida and lamina densa. Therefore, IgG autoantibodies against LM332 react with the dermal side of salt-split normal human skin [10]. A small number of patients have IgA and IgE autoantibodies that are reactive with LM332 [11]. In general, circulating autoantibodies in anti-LM332-type MMP patients may target any subunit of LM332, and autoantibodies against the LMα3 subunit might result in wider mucosae involvement [12].

Most published articles on anti-LM332-type MMP are case reports, whereas systematic reports are rare. In this review, we attempt to summarize current progress with respect to the diagnosis and treatment of anti-LM332-type MMP, as well as its potential correlations with tumors.

## 2. History of Anti-LM332-Type MMP

Anti-LM332-type MMP was formerly known as anti-epiligrin cicatricial pemphigoid (AECP) due to scarring formation in lesions [7]. AECP was first described in 1992; the patients of this disease group have autoantibodies that can bind with “epiligrin”, a constitutive protein in the epidermal basement membrane localized at the interface of the lamina lucida subregion [13]. Initial studies showed that most AECP patients had IgG autoantibodies against the LMα3 and LMγ2 subunits of LM332 [1,14]. Subsequently, epiligrin was considered to be LM332, and this hypothesis was confirmed [15]. Through various experimental studies, such as studies on passive transfer of rabbit anti-LM332 IgG antibodies to neonatal mice and the transplantation of IgG antibodies of AECP patients to server combined immune-deficient mice, the pathogenicity of anti-LM332 autoantibodies was confirmed [16,17]. Currently, this disorder is known as anti-LM332-type MMP and is distinguished from other types of MMP.

## 3. Clinical and Histological Features of Anti-LM332-Type MMP

MMP is a rare subepidermal autoimmune bullous disease. The incidence of anti-LM332-type MMP is relatively low, accounting for about 20% of MMP cases [18]. Patients with anti-LM332-type MMP are generally more than 65 years old, although there are some younger patients [19]. Anti-LM332-type MMP can last for a long time, and the frequency of occurrence is equal among men and women [20]. The involvement sites of anti-LM332-type MMP are mainly mucosae with erosive and/or vesiculobullous lesions and include ocular, nasal, pharyngeal, laryngeal, esophageal and genital mucosae [21]. Among these mucosal sites, oral mucosa is affected most frequently, followed by ocular mucosa, with other mucosal sites involved less frequently [21]. In our latest study, which included 133 anti-LM332-type MMP patients, 89% had oral lesions, 43% had ocular lesions, 19% had pharyngeal lesions, 15% had laryngeal lesions, 11% had genital lesions, 6% had nasal lesions and 3% had esophageal lesions [22]. Compared with other types of MMP, the oral and ocular mucosae were involved more commonly and seriously in patients with anti-LM332-type MMP [23]. Moreover, oral destructions may result in airway obstruction, whereas the involvement of ocular mucosa usually includes scarring; these complications may lead to mortality and blindness, respectively [24]. The presence of skin lesions is not a necessary condition for the diagnosis of anti-LM332-type MMP, although more than 60% of patients present with skin involvement [12,22,24,25].

Subepidermal blistering is the main histopathological characteristic of anti-LM332-type MMP. It is sometime impossible to obtain biopsy specimens, especially from ocular mucosa [26]. Subepidermal inflammatory infiltration is not an essential condition for diagnosis, as some patients do not present with inflammatory infiltrates of eosinophils, neutrophils or lymphocytes [1,14,20,27,28].

## 4. Diagnosis of Anti-LM332-Type MMP

The diagnosis of anti-LM332-type MMP is complicated, and it is indistinguishable from other subepidermal autoimmune blistering diseases in terms of clinical features, especially epidermolysis bullosa acquisita (EBA) [29]. Currently, there is no uniform standard for diagnosis of anti-LM332-type MMP. However, most specialists agree on a set of criteria for the diagnosis of anti-LM332-type MMP, which includes (i) subepidermal mucosal blisters, (ii) linear deposits of IgG with or without C3 on the BMZ in direct immunofluorescence (DIF), (iii) IgG autoantibodies that react with the dermal side of salt-split normal human skin and (iv) immunoblotting evidence of circulating LM332 IgG autoantibodies. Our laboratory, as well the laboratories of other researchers, has diagnosed anti-LM332-type MMP based on the following criteria: (i) IgG deposition in BMZ or IgG autoantibodies that react with the dermal side of salt-spit skin, (ii) patient serum that is reactive with LM332 and (iii) mucosal lesions [12,28,30,31]. J.B. Terra et al. offered novel insight, suggesting that a patient can be diagnosed with anti-LM332-type MMP if they satisfy three or more major criteria, namely (i) subepithelial mucosal erosions or blisters frequently associated with the scarring phenotype, (ii) n-serrated pattern IgG depositions in the BMZ detected by DIF and (iii) IgG autoantibodies that react with the dermal side of salt-split normal human skin under indirect immunofluorescence (IIF) [31]. Although these diagnosis criteria are not identical, both anti-LM332 autoantibody and mucosal lesions are very important for diagnosis of anti-LM332-type MMP. 

Detection of anti-LM332 autoantibodies in MMP patients is the most important criterion for the diagnosis of anti-LM332-type MMP. However, there is currently no established systematic detection method for anti-LM332 autoantibodies [32]. In order to better understand the detection of anti-LM332 autoantibodies, we summarized some common detection methods and their corresponding results in Table 1. 

Immunofluorescence (IF) is commonly used to detect anti-LM332 autoantibodies, including DIF, IIF and IIF, using 1M-NaCl-split normal human skin (ssIIF) [39]. The positive rates of DIF, IIF and ssIIF were in the ranges of 88–93%, 56–100% and 87–100%, respectively (Table 1). However, some anti-LM332-type MMP patients are negative according to DIF, IIF and ssIIF, possibly due to low titers of combined and circulating autoantibodies [22]. In addition, biopsy specimens are not always available, which may lead to an absence of DIF results [26]. Importantly, these IF assays are not anti-LM332 antibody-specific; therefore, the results should be interpreted with caution.

Enzyme-linked immunosorbent assay (ELISA) is another serological test method to detect anti-LM332 autoantibodies. ELISA is easy to operate with high sensitivity and has a positive rate of 78–100% (Table 1). However, there is currently no commercially available ELISA kit for detection of anti- LM332 autoantibodies [32]. Immunoblotting (IB) is a time-consuming and cumbersome method and can only be performed in a small number of laboratories [40]. Additionally, the positive rate of IB for detection of anti-LM332 autoantibodies is about 50–94% (Table 1). Both ELISA and IB are anti-LM332 antibody-specific, meaning that their results are widely accepted.

We also summarized biochip assay, n-serrated pattern fluorescence and the gold-standard detection method, immunoprecipitation (IP) of radiolabeled cultured keratinocytes, which all have 100% positive rates (Table 1). However, biochip assay and IP are not routinely used for laboratory detection of anti-LM332 autoantibodies due to the high cost and time-consuming processes, with limited literature concerning these methods, although biochip assay is a commercially available and validated detection method [35,38]. A study indicated that anti-LM332-type MMP shows an n-serrated pattern fluorescence in DIF, with a positive rate of 100% (Table 1) [31]. However, anti- LMγ1 pemphigoid (p200) also shows an n-serrated pattern fluorescence, and other serological tests are needed to distinguish them, undoubtedly decreasing the reliability of this method [31,41].

In summary, both IB and ELISA are commonly used for specific detection of anti-LM332 autoantibodies. Although ELISA is relatively simple and fast, IB is able to provide not only positivity for anti-LM332 autoantibodies but also the subunit(s) of LM332 recognized by autoantibodies in patient sera [42]. Importantly, some patient sera are only positive in IB or ELISA. Therefore, we recommend using both IB and ELISA methods to detect anti-LM332 antibodies in sera of suspected anti-LM332-type MMP patients. There is an urgent need for commercially available ELISA kits to detect antibodies against LM332. We also hope to emphasize that, in addition to IgG, IgA anti-LM332 autoantibodies should also be analyzed, as some patients have only IgA autoantibodies against LM332.

Recently, we identified a patient diagnosed with both anti-LMγ1 pemphigoid and anti-LM332-type MMP and found that the seral levels of autoantibodies against LM332 detected by ELISA in this patient changed with disease development, suggesting that detection of anti-LM332 autoantibodies has not only diagnosis value but also evaluation value regarding the treatment effect [28]. 

## 5. Treatment of Anti-LM332-Type MMP

At present, there is no standard method for the treatment of MMP. In order to provide improved treatment to MMP patients, “low-risk” and “high-risk” are often used to distinguish the severity of the disease [43]. MMP patients with only oral mucosal and/or skin lesions are usually considered to be “low-risk”; patients with one or more non-oral mucosal involvements are considered to be “high-risk” [43]. Low-risk patients are usually treated with topical corticosteroids that have mostly positive therapeutic effects, such as gels and metered-dosed inhalers [44,45,46]. Rinsing treatment with diluted betamethasone sodium phosphate tablets (0.5 mg) is often recommended for 2–3 min [45]. Corticosteroid injections should be initiated if the result of corticosteroid gels is not satisfactory, such as triamcinolone hexacetonide at a dose of 5 to 10 mg/mL every 2 to 4 weeks [46]. In addition to corticosteroids, topical therapy can be applied, such as tacrolimus [47]. T. Assmann et al. described two patients with oral mucosal pemphigoid, in which both systemic and topical corticosteroids had no significant effect; in these patients, lesions gradually decreased after 2–3 months of 0.1% tacrolimus treatment [47].

However, high-risk patients, systemic treatment may have a better effect, especially for ocular MMP [48]. Numerous systemic therapeutic options are available, including corticosteroids, azathioprine, mycophenolate mofetil, cyclophosphamide, dapsone and other sulfones; systemic corticosteroids have been recommended as first-line treatment [49,50,51]. In general, the dose of systemic corticosteroids is 0.5–1.0 mg/kg/day, whereas in severe cases, 2 mg/kg/day combine with cyclophosphamide is used, and the dose is gradually reduced when a positive effect is achieved in order to avoid other serious complications [2,45,47]. Oral prednisolone (30–60 mg/day) has been frequently used for the treatment of MMP, usually with positive therapeutic effects [3,52]. However, this treatment method generally requires a long period of time for remission and is associated with many side effects [20,53]. Systemic corticosteroids alone cannot achieve a complete cure for MMP because they are less effective on mucosal lesions and are therefore often accompanied by adjuvant immunosuppression [54]. Recent studies have shown that using immunosuppressants such as rituximab and gamma globulin as adjuvant therapy to regulate the patient’s immune system can achieve improved clinical effects and reduce the side effects of corticosteroids [55]. Conventional immunosuppressive treatment has little effect on immunocompromised patients, in which case intravenous immunoglobulins are appropriate [45]. In addition, combination with rituximab therapy can control the deterioration of MMP in a shorter period of time, requiring on one-third of the traditional treatment dose [54]. However, combination rituximab treatment is subject to some limitations, such as high cost and high recurrence rate [56]. Therefore, it is crucial to find a more effective MMP treatment method with an affordable price. For the treatment of anti-LM332-type MMP, we can currently only refer to the treatments used for MMP.

## 6. Induction Factors of Anti-LM332 Autoantibodies

There are many possible reasons for production of anti-LM332 autoantibodies, including geography, ethnicity, genetics and drug factors [21,51,57]. However, some studies have suggested that anti-LM332 autoantibody production was caused by the loss of immune tolerance to LM332 in the basement membrane, resulting in inflammation and decreased adhesion of the basement membrane [48,58,59]. One patient was reported to have been diagnosed with anti-LM332-type MMP two days after a diphtheria tetanus vaccination, suggesting that anti-LM332-type MMP might be caused by non-specific immune activation [60]. In addition, frequent relapses of bullous pemphigoid led to the production of anti-LM332 autoantibodies, possibly due to BP180 located near LM332 in the BMZ [61]. Our group also recorded two patients who produced anti-LM332 autoantibodies, one after acute mercury poisoning and the other after amalgam tooth restorations.

## 7. Anti-LM332-Type MMP and Tumors

Since Gibson et al. reported a case of anti-LM332-type MMP that coincided with lung tumor in 1997, a growing number of studies have posited the possibility that anti-LM332-type MMP patients show a higher risk for tumors, including pancreatic tumors, colon tumors, gastric tumors and lymphoma [62,63,64]. Satoko Matsushima et al. investigated 16 cases of anti-LM332-type MMP, with 5 cases (31.2%) positive for tumors [32]. In 2021, our group reported that 8 of 55 (14.5%) anti-LM332-type MMP patients developed tumors, and some patients had more than one kind of tumor [12]. We recently summarized 133 anti-LM332-type MMP cases reported between 2001 and 2014; and among them, 22 (17%) cases developed tumors, 4 of which were positive for lung tumors [22]. 

In the present study, we reviewed articles published between 1997 and 2021 and counted 344 reported cases of anti-LM332-typeMMP patients. A total of 75 (21.80%) of the 344 anti-LM332-type MMP patients presented with tumors before or after the onset of anti-LM332-type MMP. Among these 75 patients, 6 were reported to be positive for more than one kind of tumor, including 4 patients with 2 tumors, 1 patient with 3 tumors and 1 patient with 4 tumors [22,65]. Our summary provides further support for the potential relationship between anti-LM332-type MMP and tumor development.

Various tumors reported in anti-LM332-type MMP cases are summarized in Table 2. In anti-LM332-type MMP patients, tumors occur most frequently in the lungs (22.67%), followed by the gastric system (17.33%) and the uterus (13.33%) (Table 2). Based on current data, we cannot conclude whether the sites or types of tumors are specifically related to anti-LM332-type MMP [62]. 

Among these 75 patients, autoantibody information was available for 52 patients (summarized in Table 3). Circulating IgG autoantibodies against the LMα3 subunit is the most common (67.31%), followed by LMγ2 (53.85%) and LMβ3 (36.54%) (Table 3). Based on these data, we can conclude that subunit reactivities are not associated with tumor type. This is consistent with our previous conclusion, although for some patients, no information was available on the LM subunits [22]. 

Among these 75 patients, ethnic information was available for 43 patients. A total of (81.40%) patients are Asian (Japanese), whereas 6 (13.95%) are Caucasian and 2 (4.65%) are of a dark-skinned race. One possible reason for the higher tumor positivity rate among Japanese people is the HLA- DRB1*14 and HLA-DQB1*05:03 alleles, which often appear in Japanese individuals and are strongly correlated with MMP [84]. 

In these 75 patients, some patients developed tumors within 14 months after the onset of anti-LM332-type MMP [66]. The short interval indicates that the diagnosis of anti-LM332-type MMP is crucial and should be done as soon as possible. Once the diagnosis is confirmed, tumor scans are recommended, particularly in the lungs and digestive system [20,66].

Until now, the correlation mechanism between anti-LM332-type MMP and tumors has been unclear, although other studies and our summary show a possible correlation between them [28,69,76]. It is widely accepted that LM332 plays an important role in the relationship between anti-LM332-type MMP and tumors, especially in squamous cell tumors [32]. LM332 is not only a component of BMZ but also a major protein in the extracellular matrix (ECM) and is important in the processes of tumor cell proliferation, migration, infiltration and wound healing [85,86]. Studies have shown that the BMZ and ECM are the main barriers against tumor cell invasion [87,88]. However, large amounts of anti-LM332 autoantibodies of anti-LM332-type MMP patients may break the BMZ by targeting LM332 proteins. Furthermore, large amounts of LM332 proteins are produced and deposited into the ECM [89,90]. The imbalance of the ECM activates a series of signal pathways, such as the PI3-akt pathway, facilitating tumor cell migration and invasion [91]. In addition, the dysregulation of LM332 protein can promote the differentiation of tumor-associated fibroblasts, the major components of solid neoplasm, to form a microenvironment that is beneficial for angiogenesis in tumor tissues [92,93].

It remains unclear whether there is a potential correlation between the severity of mucosal involvement and tumors in anti-LM332-type MMP, although two studies found that the skin or ocular involvements disappeared after excision of gastric or cervical tumors [27,76]. Many studies have shown that tumor cells in the lungs and colon express and secrete a large amount of LM332 proteins, which may promote the production and secretion of anti-LM332 autoantibodies and result in more serious skin and mucosal membrane lesions [27,94,95]. However, it is not clear what form of LM332 is produced by aggressive tumor cells. Some studies have suggested that the cellular expression of the LMγ2 chain of LM332 is a sign of the aggressiveness of tumor cells and can be used to distinguish benign tumors from malignant tumors [94,95,96]. LMγ2 is only expressed as a subunit of LM332 on the basement membrane in normal cells, whereas it is expressed as a monomer in tumor cell cytoplasm. This has been confirmed in lung tumors, squamous cell tumors and colon tumors [33,97,98]. However, whether tumor cells can cause more serious lesions than secreted monomer LMγ2 is not fully understood.

## 8. Limitations

Among the anti-LM332-type MMP cases we reviewed for the present study, some cases might have been counted twice. It is possible that some of the cases may have been reported as an independent report and also summarized in an article detailing multiple cases. In addition, the incomplete information of some patients may have influenced the accuracy of the summarized data.

## 9. Conclusions

Anti-LM332-type MMP is a rare subepidermal autoimmune bullous disease, and the detection of autoantibodies against LM332 is the basis for the diagnosis and further treatment of anti-LM332-type MMP. The use of ELISA for detection of autoantibodies against LM332 is a relatively promising diagnosis method for future clinical application. With the increase in diagnosis ability and in-depth study of the pathogenic mechanism of anti-LM332-type MMP, a superior treatment method may be developed, and the correlations of tumor development with anti-LM332-type MMP might also be clarified in the future. 

## Figures and Tables

**Table 1 biomolecules-12-01461-t001:** Immunological methods for detection of anti-LM332 autoantibodies.

Method	Positive Rate	References
IB	50–94%	[7,24,31,33,34]
ELISA	78–100%	[7,31,34,35]
DIF	88–93%	[7,22,24]
IIF	56–100%	[22,24,31]
ssIIF	87–100%	[7,22,31,36,37]
IP	100%	[24,33]
N-serrated pattern	100%	[31]
Biochip assay	77–96%	[37,38]
Footprint assay	100%	[38]

IB, immunoblotting; ELISA, enzyme-linked immunosorbent assay; DIF, direct immunofluorescence; IIF, indirect immunofluorescence; ssIIF, IIF using 1M-NaCl-split normal human skin; IP, immunoprecipitation.

**Table 2 biomolecules-12-01461-t002:** The types of tumors in 75 anti-LM332-type MMP patients.

Tumor Type	Number of Patients (%)	References
Lung	17 (22.67%)	[11,12,22,24,26,31,32,63,66,67,68]
Gastric	13 (17.33%)	[22,24,27,39,66,69,70,71]
Uterine	10 (13.33%)	[22,24,66,69,72,73]
Pancreatic	6 (8.00%)	[22,40,74,75]
Colon	6 (8.00%)	[22,24,66,72]
Ovary	5 (6.67%)	[18,22,76,77,78]
Prostate	4 (5.33%)	[20,22,79]
Thyroid	4 (5.33%)	[22,65]
B-cell lymphoma	3 (4.00%)	[22,80]
Leukemia	3 (4.00%)	[22,70]
Liver	3 (4.00%)	[22,39,81]
Kidney	3 (4.00%)	[22,65,82]
Adenocarcinoma	2 (2.67%)	[22,83]
Tongue	2 (2.67%)	[79]
Pharynx	1 (1.33%)	[20,22,65]
Breast	1 (1.33%)	[22]
Cutaneous lymphoma	1 (1.33%)	[62]
Total	75 (100%)	/

**Table 3 biomolecules-12-01461-t003:** The tumor types and LM332 subunits recognized in patient sera among 52 anti-LM332-type MMP cases.

Tumor (Number of Patients)	LM332 Subunits, Positive Cases (%)		
	LMα3	LMβ3	LMγ2
Lung (17)	10 (58.82%)	4 (23.53%)	7 (41.18%)
Gastric (13)	7 (53.85%)	4 (30.77%)	3 (23.08%)
Uterine (10)	4 (40.00%)	unknown	2 (20.00%)
Pancreatic (6)	3 (50.00%)	3 (50.00%)	4 (66.67%)
Colon (6)	4 (66.67%)	unknown	unknown
Ovary (5)	unknown	unknown	4 (80%)
Prostate (4)	1 (25.00%)	2 (50.00%)	1 (25%)
Thyroid (4)	1 (25.00%)	0 (0.00%)	4 (100.00%)
B-cell lymphoma (3)	1 (33.33%)	unknown	1 (33.33%)
Leukemia (3)	unknown	2 (66.67%)	unknown
Liver (3)	2 (66.67%)	2 (66.67%)	2 (67.67%)
kidney (3)	unknown	unknown	2 (67.67%)
Adenocarcinoma (2)	2 (100.00%)	1 (50.00%)	1 (50.00%)
Tongue (2)	1 (50.00%)	1 (50.00%)	2 (100.00%)
Pharynx (1)	1 (100.00%)	1 (100.00%)	1 (100.00%)
Breast (1)	0 (0.00%)	0 (0.00%)	1 (100.00%)
Cutaneous lymphoma (1)	1 (100.00%)	0 (0.00%)	0 (0.00%)
Total	35 (67.31%)	19 (36.54%)	28 (53.85%)

## Data Availability

No new data were created or analyzed in this study. Data sharing is not applicable to this article.

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
