# Peer review of "Anti-Laminin 332-Type Mucous Membrane Pemphigoid"

_biomolecules, 2022, doi:10.3390/biom12101461_

Round 1
Reviewer 1 Report
Patients with only oral mucosa and/or skin lesions are usually 155 considered to be “low-risk”; patients with one or more mucosal involvements are considered to be “high-risk” [39].
Could you please explain this sentence? Patients with only oral mucosa are considered to be low risk. Patient with one or more mucosa involved are considered to be high risk. I don't understand this. Oral mucosa is one mucosa, so is that low risk or high risk?
I had a tough time understanding this paper, since there are issues with syntax, grammar, and choice of words. Also, I didn't understand the point of the paper. Is it to introduce a new test to detect anti-LM332 antibodies or is it to review all the previous cases of this entity?
Author Response
The authors’ replies were shown with red-colored words following each comment of the reviewers.
Comments of Reviewer 1
- Patients with only oral mucosa and/or skin lesions are usually 155 considered to be “low-risk”; patients with one or more mucosal involvements are considered to be “high-risk” [39]. Could you please explain this sentence? Patients with only oral mucosa are considered to be low risk. Patient with one or more mucosa involved are considered to be high risk. I don't understand this. Oral mucosa is one mucosa, so is that low risk or high risk?
Reply: Thank you for your kind notification. To make it clearly, we have revised this description in the line162-164 of the revised manuscript, which is also shown below.
“MMP patients with only oral mucosa and/or skin lesions are usually considered to be “low-risk”; patients with one or more non-oralmucosae involvements are considered to be “high-risk” [39].”
(2) I had a tough time understanding this paper, since there are issues with syntax, grammar, and choice of words. Also, I didn't understand the point of the paper. Is it to introduce a new test to detect anti-LM332 antibodies or is it to review all the previous cases of this entity?
Reply: Thank you for your kind suggestion. This newly submitted manuscript has been widely revised on English by a professional language editor.
The purpose of our manuscript is not to introduce a new test for detection of anti-LM332 autoantibody, but to summarize all previously reported cases of anti-LM332 MMP and focus on laboratory diagnostic methods, and the potential correlations between anti-LM332 MMP and tumors.
Reviewer 2 Report
In this paper authors offered a complete and extensive review regarding clinical diagnostic and therapeutic aspects of laminin 332 mucous membrane pemphigoid.
The work is well structured and interesting for clinicians in the field.
Nevertheless, I found several spelling errors (some of them listed in the following lines) which penalizes the work and makes it less readable and appealing. Therefore, I suggest a moderate English spelling improvement by a native English editor.
Here some examples:
line 15: On treatment- " concerning treatment" is more appropriate
line 28: autoimmunity should be changed into autoimmune
line 31: zone insted of zoon
line 45: change into subunit instead of subunits
line 65: of MMP it's not clear. It should be specified that it refers to MMP patients
line 67: last with. "with" in this pharse is not clear
line 67: cocurrence should be changed into occurence
line 73: repetition of patients
lines 83-86 : the meaning of this sentence is not clear
lines 259 prudction istead of production
line 260: "result in" instead of result alone
In general the work in its content is complete and well structured. However, I suggest some reccomendations to improve the work:
- At line 118 authors declare that " some anti-LM332-type MMP patients are negative in DIF, IIF and ssIIF". This is true , but authors should also include some speculations or hypothesis to explain this trend (ie. low titers of circulating antibodies in MMP)
- In the section Diagnosis authors should emphasize which tests are commercially available and validated ( i.e BIOCHIP). This would help clinicians without specialized laboratorories. At the same time authors should emphasize the urgent need for the developement of commercially available ELISA detecting antibodies against laminin 332.
- In section Treatment I suggest including recent guidelines among references PMID: 34309078
In section Anti-LM332-type MMP and tumor authors correctly declare that after diagnosis tumor scans are recommended. It would be interesenting if authors described also their protocol if the have one ( i.e total body scan rather than limited to selected areas ?) discussing it with different approaches in literature if present.
Author Response
The authors’ replies were shown with red-colored words following each comment of the reviewers.
Comments of Reviewer 2
In this paper authors offered a complete and extensive review regarding clinical diagnostic and therapeutic aspects of laminin 332 mucous membrane pemphigoid.
The work is well structured and interesting for clinicians in the field.
Nevertheless, I found several spelling errors (some of them listed in the following lines) which penalizes the work and makes it less readable and appealing. Therefore, I suggest a moderate English spelling improvement by a native English editor.
Reply: Thank you for your kind suggestion. This newly submitted manuscript has been widely revised on English by a professional language editor.
Here some examples:
line 15: On treatment- " concerning treatment" is more appropriate
Reply: we have replaced "on treatment" by “concerning treatment” in new line 16.
line 28: autoimmunity should be changed into autoimmune
Reply: we have replaced "autoimmunity" by “autoimmune” in new line 29.
line 31: zone insted of zoon
Reply: we have replaced "zoon" by “zone” in new line 32.
line 45: change into subunit instead of subunits
Reply: we have replaced "subunits" by “subunit” in new line 46.
line 65: of MMP it's not clear. It should be specified that it refers to MMP patients
Reply: We have replaced “MMP” by “MMP patients” in new line 67.
line 67: last with. "with" in this pharse is not clear
Reply: We have replaced “with” by “for” in new line 69.
line 67: cocurrence should be changed into occurrence
Reply: we have replaced "cocurrence" by “occurrence” in new line 70.
line 73: repetition of patients
Reply: Thank you very much. We have deleted the first “patients” in new line 75.
lines 83-86 : the meaning of this sentence is not clear
Reply: We have revised the sentence in line 83-86 (currently lines 86-88), which is also shown below.
“Subepidermal inflammatory infiltration is not an essential condition for diagnosis, because some patients do not have inflammatory infiltrates of eosinophils, neutrophils or lymphocytes”.
lines 259 prudction istead of production
Reply: we have replaced "prudction" by "production" in new line 269.
line 260: "result in" instead of result alone
Reply: we have replaced "result" by “result in” in new line 270.
In general the work in its content is complete and well structured. However, I suggest some reccomendations to improve the work:
- At line 118 authors declare that " some anti-LM332-type MMP patients are negative in DIF, IIFand ssIIF". This is true , but authors should also include some speculations or hypothesis to explain this trend (ie. low titers of circulating antibodies in MMP)
Reply: As suggested, we revised this sentence by adding some speculations in new line 121-123, which is also shown below.
“However, some anti-LM332-type MMP patients are negative in DIF, IIF and ssIIF, possibly due to low titers of combined and circulating autoantibodies [22]”.
- In the section Diagnosis authors should emphasize which tests are commercially available and validated ( i.e BIOCHIP). This would help clinicians without specialized laboratorories. At the same time authors should emphasize the urgent need for the developement of commercially available ELISA detecting antibodies against laminin 332.
Reply: In the revised manuscript, we have clearly described that biochip assay is commercially available and validated, and also emphasize it is urgent need for the commercially available ELISA to detect autoantibodies against LM332. The revisions were shown in new line 139 and 150-151 of the revised manuscript.
- In section Treatment I suggest including recent guidelines among references PMID: 34309078
Reply: We added corresponding treatment according to references PMID: 34309078 (current reference 41) in new line 164-167 and line 189-190, which were also shown below.
Line 164-167:
“Low-risk patients are usually treated with topical corticosteroids and have a good therapeutic effect mostly, such as gels and metered-dosed inhalers [40-42]. While rinsing treatment is often with diluted betamethasone sodium phosphate tablets (0.5mg) for 2-3min [41].”
Line 189-190:
Conventional immunosuppressive treatment has little effect on immunocompromised patients and intravenous immunoglobulins are appropriate to them [41].
In section Anti-LM332-type MMP and tumor authors correctly declare that after diagnosis tumor scans are recommended. It would be interesenting if authors described also their protocol if the have one ( i.e total body scan rather than limited to selected areas ?) discussing it with different approaches in literature if present.
Reply: There are no studies describing about the protocol for tumor scanning, but we recommend to at least scan lung and digestive system according to our summaries. And we added “particularly in lung and digestive system” in new line 248-249.
Round 2
Reviewer 1 Report
Minor English language edits needed.
Author Response
Reply: Thanks, the English Editing was completed in the proof stage.